# A High Sensitivity Custom-Built Vibrating Sample Magnetometer

**Jared Paul Phillips** [1,*] ![], **Saeed Yazdani** [1] ![], **Wyatt Highland** [2] **and Ruihua Cheng** [1]

[1] Department of Physics, Indiana University Purdue University Indianapolis (IUPUI), Indianapolis, IN 46202, USA; syazdani@iupui.edu (S.Y.); rucheng@iupui.edu (R.C.)
[2] Department of Mechanical and Energy Engineering, Purdue School of Engineering, Indianapolis, IN 46202, USA; whighlan@iu.edu
[*] Correspondence: japphill@iu.edu

**Abstract:** This work details the construction and optimization of a fully automated, custom-built, remote controlled vibrating sample magnetometer for use in spintronics related research and teaching. Following calibration by a standard 6 mm diameter Ni disc sample with known magnetic moment, hysteresis measurements of Nd-Fe-B thin films acquired by this built vibrating sample magnetometer were compared to the data taken using a commercial superconducting quantum interference device and showed very similar results. In plane and out of plane magnetic hysteresis data acquired for 25 nm Fe thin films are also presented. The developed vibrating sample magnetometer is able to achieve a sensitivity approaching $1 \times 10^{-5}$ emu. Further alterations to the design that may improve beyond this limit are also discussed.

**Keywords:** instrumentation; magnetometry; vibrating sample magnetometer; spintronics; thin films; hysteresis loop; magnetic moment; ferromagnetism

## 1. Introduction

Magnetic materials are crucial elements in a vast array of modern devices. This trend is likely to continue in the foreseeable future as the field of spintronics advances, with research devoted to technological applications that incorporate magnetic materials at the nanoscale in increasingly more complex ways [1–3]. In most experimental scenarios, the magnetic material of interest is present in trace amounts or in the form of a thin film, which necessitates the use of a highly sensitive magnetometer [4]. There are various types of magnetometers, with three of the most common being the magneto-optic Kerr effect magnetometer (MOKE), the superconducting quantum interference device (SQUID), and the vibrating sample magnetometer (VSM). A MOKE functions based on the Kerr effect [5] and is well suited for probing the surface of thin films. A SQUID offers highly sensitive measurements on the order of $1 \times 10^{-8}$ emu [6], however, the materials required to construct and operate a SQUID can be prohibitively expensive. In contrast, the VSM, first developed by Foner in 1959 [7], is an instrument widely used in both research and teaching labs, due in large part to the ruggedness and simplicity of its design, coupled with the precision and repeatability of its measurement technique. Although a commercial VSM can provide sensitivities in the range of $1 \times 10^{-7}$ emu, a much more cost-effective and easily adaptable option can often be built in house to address specific needs [8–12].

In this work, the construction of a custom-built and highly sensitive VSM is described at length. First, we discuss the physical concepts and principles on which the VSM is based. Then, we detail the construction of the device, listing the components used, followed by the steps involved in calibrating and optimizing the VSM. Next, data acquired from the VSM are compared to results obtained from a commercial SQUID for the same thin film samples. Finally, we present data acquired from a 25 nm Fe thin film that further verify the VSM accuracy and gauges its sensitivity.

## 2. VSM Operating Principle

A VSM functions based on Faraday's law. Specifically, the sinusoidal motion of a magnetized sample generates an electromagnetic field (EMF) in a set of pick-up coils. For samples with small dimensions relative to the size of the pickup coils, the sample essentially behaves like a miniature magnet that generates a magnetic field in space (Figure 1), given by the dipole approximation:

$$B(r) = \frac{\mu_0}{4\pi} \left[ \frac{3r(\boldsymbol{m} \cdot \boldsymbol{r})}{r^5} - \frac{\boldsymbol{m}}{r^3} \right] , \tag{1}$$

where $\boldsymbol{m}$ is the magnetic moment of the sample.

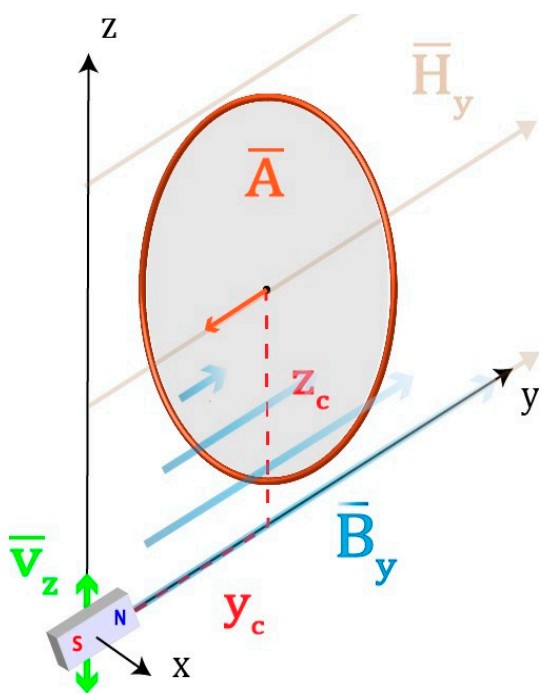

**Figure 1.** The y-component of magnetic flux, $B_y$, for a dipole in a uniform magnetic field, $H_y$, passing through a single turn of a pickup coil of area, A, centered on the x axis; $v_z$ indicates the direction of oscillation of the dipole.

The flux through a single loop of conducting wire is

$$\Phi = \int_A B(r) \cdot dA, \tag{2}$$

and an induced voltage, $V_{ind}$, is created in the coil due to an oscillation of the sample equal in magnitude to $d\Phi/dt$. This is the quantity that is measured experimentally by a VSM setup through the accompanying electronics.

An alternative way of representing $V_{ind}$ for the purpose of VSM design can be done through the theory of reciprocity, which states that the mutual inductance linking two coils is independent of which one carries the current:

$$\frac{\Phi_{21}}{I_1} = \frac{\Phi_{12}}{I_2} , \tag{3}$$

Here, $\Phi_{21}$ is the flux through coil 2 due to the field generated by coil 1 and vice versa. If coil 1 is replaced by a dipole with overall moment m = $I_1 dA_1$, the equality can be rewritten. After substituting, the flux through coil 2 (one turn of the sensor coil) is given by

$$\Phi_{21} = \frac{B_2 m}{I_2} \, , \qquad (4)$$

where $B_2$ is the "B-field" in the y-direction at the surface of the sample due to a "fictitious" current in the coils, $I_2$, and $m$ is the magnetic moment of the sample. When the magnetic moment is oscillated in the z direction, the $V_{ind}$ is then

$$\frac{d\Phi_{21}}{dt} = \frac{d\Phi_{21}}{dz}\frac{dz}{dt} = v_z \frac{d\left(\frac{B_2 m}{I_2}\right)}{dz} = m v_z S(r) \qquad (5)$$

where $v_z$ is the user specified velocity of the sample in the form of $A\omega\cos(\omega t)$. The $S(r)$ term is known as the sensitivity function and depends on the geometry of the sensor coils and their configuration in space with respect to the sample under test [13,14]. This is the parameter one seeks to maximize in the design process of the VSM.

## 3. Instrumentation Development

The base frame of our low cost VSM was built with 2 × 4" wood using glue and brass screws in order to minimize any undesired signal. A 9" diameter speaker is attached to a thick acrylic sheet, with a hole cut from its center, which is affixed to the top of the wooden frame. Super glued onto the speaker diaphragm is an acrylic disc with a threaded hole in its center, where one end of a 32" fiberglass threaded sample rod is fixed. A key feature of this design that differs from many other custom VSM setups is the use of a much longer sample rod, which was done in an attempt to reduce any unwanted mechanical or magnetic signal in the sensor coils due to close proximity of the speaker. The entire device rests on top of a large rubber mat that helps to reduce mechanical vibration. On the other end of the rod, a threaded Delrin cylinder is attached. The cylinder's function is twofold. It acts as a weight, which, in principle, helps minimize the lateral motion of the sample rod; in addition, the open end of the cylinder allows for the sample holder to be attached and detached easily, while also allowing for fine adjustments of the sample height with respect to the sensor coils. A schematic diagram of the VSM is shown in Figure 2.

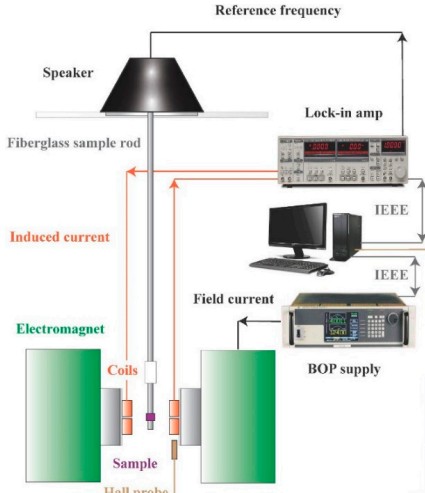

**Figure 2.** A schematic of the constructed VSM. The bipolar power supply (BOP) ramps the magnetizing field, while the lock-in amplifier drives the speaker, with attached sample rod, causing it to oscillate. This creates a time varying flux in the sensor coils, which generates a current. This sinusoidal current is read by the lock-in amplifier and the data are fed into the computer where they are recorded and plotted via LabVIEW (2021, National Instruments, Austin, TX, USA).

Each sensor coil consists of approximately 4700 turns of 42 AWG (0.0026″ diameter) copper wire wound on a 27.4 outer diameter (OD)/15 inner diameter (ID) mm bobbin. Each pair is separated by 1.2 mm in the z-direction and 20 mm in the y-direction and enclosed within a 3D printed mount, as pictured in Figure 3a. The coils are wired together in the Mallinson configuration [15], illustrated in Figure 3b. The benefits of this configuration are such that the $V_{ind}$ of all four coils add together when the changing flux originates from the space between them (i.e., from the sample). In addition, for a changing flux that originates from outside the region between the coils, the undesired EMF in the coils is attenuated.

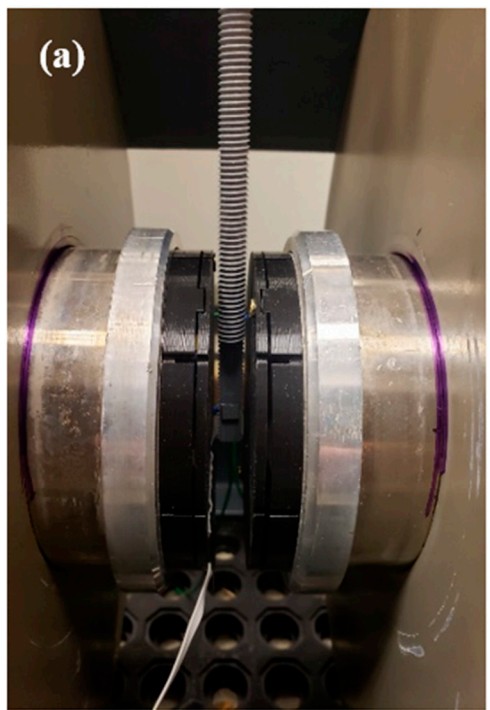 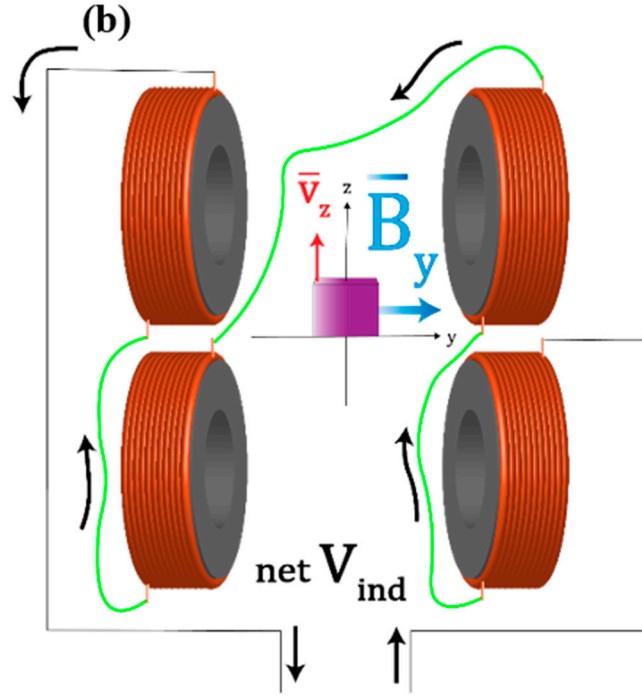

**Figure 3.** (**a**) A closeup picture of the sample holder and the coils shielded by 3D printed covering. (**b**) A net $V_{ind}$ is generated in the coils due to the oscillation of the sample in the z direction.

A 1 kW Kepco bipolar power supply (BOP) is used to ramp the field of the electromagnet between ±5000 G, and a Stanford Instruments SR830 lock-in amplifier provides a sinusoidal signal to drive the speaker and reads the output from the sensor coils. To achieve optimal signal output from all four sensor coils, the corresponding values required for the lock-in output voltage (5 V) and frequency (112.91 Hz) were determined experimentally. Additional values determined via experimentation were the minimum required lock-in time constant (1 s) and wait time between data points to allow the lock-in to sufficiently settle (14 s). The BOP and lock-in control are fully automated through a GPIB interface using custom written LabView code. In addition to the standard magnetic measurement capability of a VSM, ours can be easily modified to allow for such things as temperature dependent magnetoresistance measurements, dielectric spectroscopy, and MOKE studies.

## 4. Calibration Results and Discussion

After the system was built, the VSM was carefully calibrated using two methods. In the first method, a 6 mm diameter, 32 mg nickel disc standard sample from the National Institute of Standards and Technology (NIST) was used to determine the magnetic moment calibration constant. The standard disc has a specific magnetization at a given applied field. In this case, it is 1.753 emu at 5000 Oe. To convert the raw data collected from the lock-in output to units of emu, one finds the ratio of the lock-in value in volts at 5000 Oe with 1.753 emu. Using this method, the conversion constant for the VSM was found to be 4677 emu/V. Additionally, data acquired from the VSM were compared to SQUID

data taken for the same thin film samples prepared via sputtering under high vacuum. A VSM hysteresis run of a clean Si substrate, approximately the same dimensions as the thin film substrates, was performed in order to remove the background from the raw data. This data subtraction step is necessary because Si substrates, which the thin films were deposited on, are diamagnetic in nature and contribute a negative slope to the overall signal. Following background subtraction, the data were converted to emu units and compared with the SQUID data, which can be seen in Figure 4. Here the magnetic hysteresis data in Figure 4a, obtained from SQUID and VSM, show some constraining effects in the low magnetic field region. This is because the sample, prepared under specific conditions, possesses phases other than the desired $Nd_2Fe_{14}B$ phase, and these different phases present antiferromagnetic-like interaction. In the regions where the samples are nearly saturated, at fields greater than $\pm 2500$ Oe, the difference between the data acquired by the SQUID and that of the VSM varies between 1.5% and 3%.

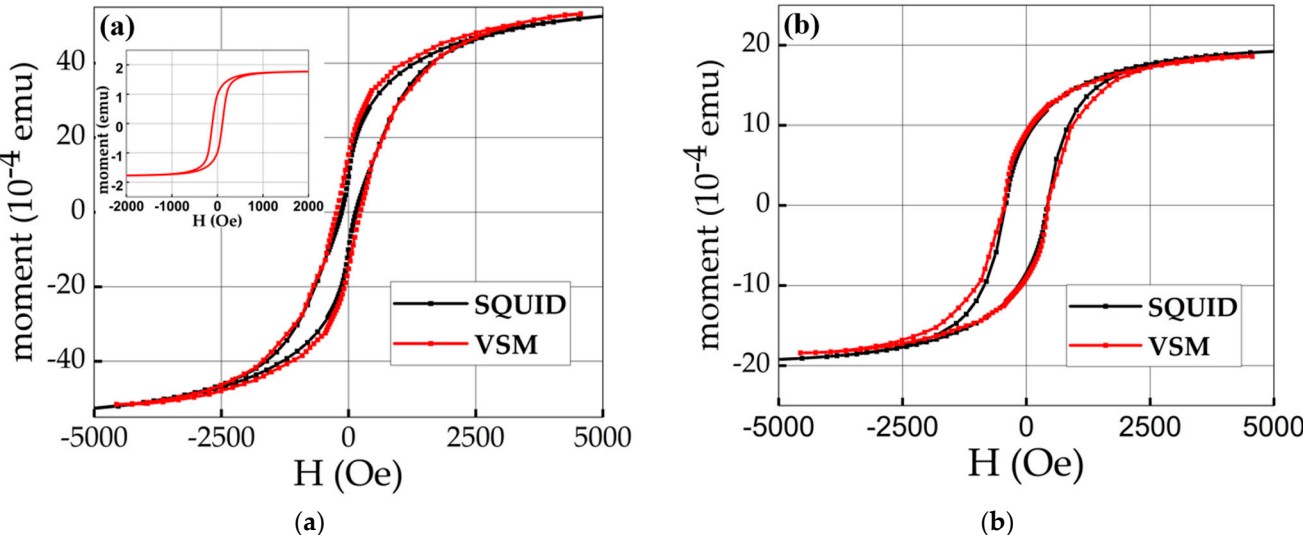

**Figure 4.** (**a**,**b**) show the comparison of magnetic moment data acquired via a commercial SQUID and the constructed VSM for two separate Nd-Fe-B thin films fabricated under different conditions. The inset of (**a**) is a hysteresis plot of the Ni disc calibration sample measured by the VSM.

After verifying the calibration of the VSM, 25 nm Fe thin films sputtered on Si substrates were prepared. In plane and out of plane hysteresis measurements were then performed on the thin films. The results, after background subtraction, are shown in Figure 5 and are in reasonable agreement with the theoretical value [16]. Approximating the surface area of the measured sample to be 25 $mm^2$, the total mass of Fe deposited on the substrate would be around 3.1 nanograms. Multiplying this by the reported saturation value of pure Fe, which is equal to 221.71 emu/g, the calculated emu equates to approximately $7 \times 10^{-4}$ emu. Comparing this theoretical value to the value measured by the VSM results in a percent difference of less than 7% between the two.

Judging from the data, it is apparent that the sensitivity of the VSM is nearly on the order of $1 \times 10^{-5}$ emu, which compares favorably to other custom VSM builds with sensitivities typically ranging from $1 \times 10^{-2}$ to $1 \times 10^{-4}$ emu [2,17,18]. This is rather promising when considering that there are several potential improvements that can be made to the existing setup. One major factor that likely has an effect on the data is thermal drift caused by the electromagnet, which is currently not water-cooled and becomes noticeably warm. When performing very small signal VSM measurements, as with background subtractions, the data shift slightly with each hysteresis loop iteration. Another source of undesired signal that might be addressed is in the wiring between the sensor coils and the lock-in. Due to the small voltages involved in a data measurement, proper wiring between the sensor coils and lock-in is crucial for optimizing the signal to noise (S/N)

ratio. For instance, in the early stages of calibrating and troubleshooting the VSM, it was discovered that a significant source of noise and abnormality in the measurements was due to insufficient shielding and isolation of the wiring. More specifically, the observed irregularities in induced voltage were likely due to slight capacitance variations building up along the length of the wires. Some steps taken that improved overall performance included shortening the length of the wires and distancing them from each other, the speaker, and the electromagnet. The wires were also secured and prevented from hanging freely, which helped to reduce mechanical vibration. Further steps to improve S/N could include adding a pair of reference coils and a reference sample, with a well-defined magnetic moment, further up the sample rod [7]. Because the reference and the sample under test would both be driven by the same rod, the amplitude and phase of the voltages they induce in their respective coils would be related. The reference output could then be used to normalize the data, as it would detect any abnormalities in the signal. This would make measurements less susceptible to stray mechanical and electrical noise and to variations in things such as vibration amplitude and frequency.

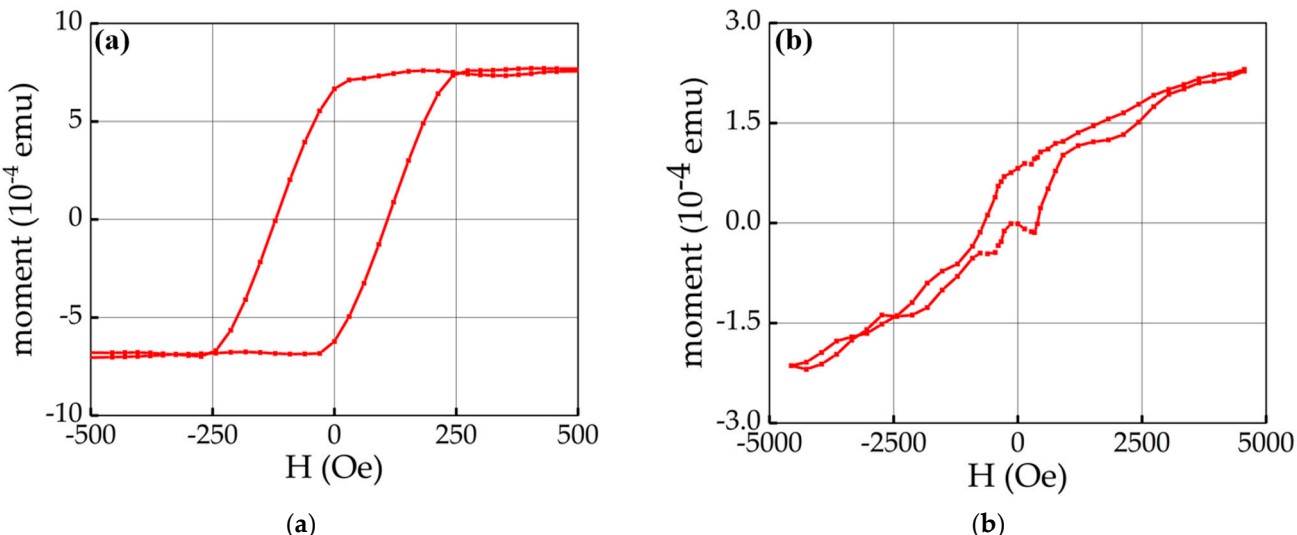

**Figure 5.** Magnetization data of a 25 nm Fe thin film sputtered on a Si substrate. (**a**) In plane, parallel to the easy axis. (**b**) Out of plane, perpendicular to the easy axis.

In addition to being a research apparatus, this developed VSM has been implemented into an advanced physics laboratory course for undergraduate students. The straightforward and user-friendly design of the VSM and the custom written LabView code that controls it gives students the opportunity for hands-on experience in experimental research studying the magnetic properties of thin films.

## 5. Conclusions

Analyzing the characteristics of nanoscale materials typically requires the use of sophisticated measuring systems; however, such instruments are not always readily available due to their cost or complexity, as is the case with a SQUID or commercial VSM. As discussed in this work, it is possible to build a highly sensitive custom-built VSM using inexpensive materials. The constructed VSM was calibrated using two methods. Initially, a Ni disc with a well-defined saturation magnetization was used to determine the calibration constant of the lock-in amplifier, followed by the comparison of hysteresis loops acquired by the VSM and a commercial SQUID for the same thin film samples. Additional measurements of 25 nm Fe thin films highlighted the sensitivity of the device, which is currently on the order of $1 \times 10^{-5}$ emu.

**Author Contributions:** Conceptualization, J.P.P., S.Y. and W.H.; methodology, J.P.P., S.Y. and R.C.; construction, J.P.P., S.Y. and W.H.; software, J.P.P. and W.H.; testing and calibration, J.P.P., S.Y. and R.C.; data analysis, J.P.P. and S.Y.; original draft preparation, J.P.P.; writing—review and editing, J.P.P., S.Y. and R.C.; supervision, R.C.; funding acquisition, R.C. All authors have read and agreed to the published version of the manuscript.

**Funding:** This research was supported by the National Science Foundation through NSF-DMR 2003057.

**Institutional Review Board Statement:** Not applicable.

**Informed Consent Statement:** Not applicable.

**Data Availability Statement:** Not applicable.

**Conflicts of Interest:** The authors declare no conflict of interest.

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
