# Peer review of "A High Sensitivity Custom-Built Vibrating Sample Magnetometer"

_magnetochemistry, doi:10.3390/magnetochemistry8080084_

Round 1
Reviewer 1 Report
It is considered for the authors choice to make more attention on describing how the setup can be improved in terms to get a better signal. And how the current signal is affected by the setup characteristics.
Author Response
Reviewer Comments:
Point 1: It is considered for the authors choice to make more attention on describing how the setup can be improved in terms to get a better signal. And how the current signal is affected by the setup characteristics.
Response 1:
Thank you for taking the time to review our article and thank you for your suggestions. To address your suggestions; we feel that the addition of a reference sample might improve the data. Reference samples are utilized in many VSM systems, including in the original design, pioneered by Simon Foner.
In Section 4, page 6, (line 266-273) we have added: “Further steps to improve S/N could include adding a pair of reference coils and a reference sample, with a well-defined magnetic moment, further up the sample rod [7]. Since the reference and the sample under test would both be driven by the same rod, the amplitude and phase of the voltages they induce in their respective coils would be related. The reference output could then be used to normalize the data, as it would detect any abnormalities in the signal. This would make measurements less susceptible to stray mechanical and electrical noise and to variations in things such as vibration amplitude and frequency.”
Reviewer 2 Report
The work by Phillips et al. reports a custom-built magnetometer with good sensitivity. The as-prepared magnetometer is potentially useful for some applications such as in research and teaching. Overall I am satisfied with the organization and presentation of the manuscript. To further improve the technical quality, the following comments could be considered.
1) It is necessary to compare this work with other custom-built magnetometers and to highlight the novelty of this work.
2) It would be better to give some comments on the constraining effect M-H loop (which is significant in the SQUIP result) as shown in Fig. 4(a).
Author Response
Reviewer Comments:
The work by Phillips et al. reports a custom-built magnetometer with good sensitivity. The as-prepared magnetometer is potentially useful for some applications such as in research and teaching. Overall I am satisfied with the organization and presentation of the manuscript. To further improve the technical quality, the following comments could be considered.
Point 1: It is necessary to compare this work with other custom-built magnetometers and to highlight the novelty of this work.
Thank you for your comments and suggestions.
Response 1: To address your first suggestion, we compared the sensitivity of our VSM to some similar custom builds, which included the addition of two more references. We have added a comparison to the manuscript. In Section 4, to the first sentence of page 6 (line 249-250), we have modified the sentence to read:
“Judging from the data, it is apparent that the sensitivity of the VSM is nearly on the order of 1×10−5 emu, which compares favorably to other custom VSM builds with sensitivities typically ranging from 1×10-2 to 1×10-4 emu [17,18,2].”
Point 2: It would be better to give some comments on the constraining effect M-H loop (which is significant in the SQUID result) as shown in Fig. 4(a).
Response 2: To address your second suggestion, we further elaborated on the hysteresis data presented in Figure 4, on page 5, section 4 (line 191-195), by adding:
“Here the magnetic hysteresis data in Figure 4a, obtained from SQUID and VSM, show some constraining effects in the low magnetic field region. This is because the sample, prepared under specific conditions, possesses phases other than the desired Nd2Fe14B phase and these different phases present antiferromagnetic-like interaction.”
Reviewer 3 Report
1. The following references should be added:
[1] B. C. Dodrill, J. R Lindemuth, Vibrating Sample Magnetometry, in Magnetic Measurement Techniques for Materials Characterization (edited by: V. Franco, B Dodrill), Springer Nature, 2021.
[2] A. Niazi, P, Poddar, A. K. Rastogi, A Precision, Low-cost Vibrating Sample Magnetometer, Current Science, 79, 1, 2000.
[3] R. M. El-Alaily, M. K. El-Nimr, S. A. Saafan, M. M. Kamel, T. M. Meaz, S. T. Assar, Construction and Calibration of a Low-cost and Fully Automated Vibrating Sample Magnetometer, J. Mag. Mag. Mat., 386, 25, 2015.
[4] V. Dominguez, A. Quesada, J. C. Mainuez, L. Moreno. M. Lere, J. Spottorno, F. Giacomone, J. F. Ferandez, A. Hernando, M. A. Garcia, A Simple Vibrating Sample Magnetometer for Macroscopic Samples, Rev. Sci. Inst., 89, 034707, 2018.
[5] V. I Nizhankovskii, L. B. Lugansky, Vibrating Sample Magnetometer with a Step Motor, Meas. Sci. Tech., 1533, 2006.
2. page 4, line 170-172: "One potential advantage of this VSM over 170 that of a commercial version is its open space and modular design, which allows it to be 171 easily adapted to our specific research needs."
This is not a valid statement, there are commercial VSMs (e.g., www.lakeshore.com) that provide an "open space" and that are of a "modular design".
Author Response
Reviewer Comments:
Point 1: The following references should be added:
[1] B. C. Dodrill, J. R Lindemuth, Vibrating Sample Magnetometry, in Magnetic Measurement Techniques for Materials Characterization (edited by: V. Franco, B Dodrill), Springer Nature, 2021.
[2] A. Niazi, P, Poddar, A. K. Rastogi, A Precision, Low-cost Vibrating Sample Magnetometer, Current Science, 79, 1, 2000.
[3] R. M. El-Alaily, M. K. El-Nimr, S. A. Saafan, M. M. Kamel, T. M. Meaz, S. T. Assar, Construction and Calibration of a Low-cost and Fully Automated Vibrating Sample Magnetometer, J. Mag. Mag. Mat., 386, 25, 2015.
[4] V. Dominguez, A. Quesada, J. C. Mainuez, L. Moreno. M. Lere, J. Spottorno, F. Giacomone, J. F. Ferandez, A. Hernando, M. A. Garcia, A Simple Vibrating Sample Magnetometer for Macroscopic Samples, Rev. Sci. Inst., 89, 034707, 2018.
[5] V. I Nizhankovskii, L. B. Lugansky, Vibrating Sample Magnetometer with a Step Motor, Meas. Sci. Tech., 1533, 2006.
Response 1: Thank you for your comments and suggestions. Regarding your first suggestion, we have added all of the references. In addition, we have cited two of the suggested references in Section 4, the first sentence of page 6 (line 249-250), where we have modified the sentence to read:
“Judging from the data, it is apparent that the sensitivity of the VSM is nearly on the order of 1×10−5 emu, which compares favorably to other custom VSM builds with sensitivities typically ranging from 1×10-2 to 1×10-4 emu [17,18,2].”
Point 2: page 4, line 170-172: "One potential advantage of this VSM over 170 that of a commercial version is its open space and modular design, which allows it to be 171 easily adapted to our specific research needs."
This is not a valid statement, there are commercial VSMs (e.g., www.lakeshore.com) that provide an "open space" and that are of a "modular design".
Response 2: To address your second suggestion, we have removed the statement: “One potential advantage of this VSM over that of a commercial version is its open space and modular design, which allows it to be easily adapted to our specific research needs.”